# Long Non-Coding RNAs as Emerging Targets in Lung Cancer

**DOI:** 10.3390/cancers15123135

**Published:** 2023-06-10

**Authors:** Jovanka Gencel-Augusto, Wei Wu, Trever G. Bivona

**Affiliations:** 1Department of Medicine, University of California San Francisco (UCSF), San Francisco, CA 94158, USA; jovanka.gencelaugusto@ucsf.edu; 2UCSF Hellen Diller Comprehensive Cancer Center, San Francisco, CA 94158, USA; 3Chan-Zuckerberg Biohub, San Francisco, CA 94158, USA

**Keywords:** LncRNAs, lung cancer, metastasis, therapy resistance, biomarkers, alternative therapies

## Abstract

**Simple Summary:**

Long non-coding RNAs (LncRNAs) are non-protein coding molecules longer than 200 nucleotides. They play essential roles in normal cell function and development, and can contribute to diseases such as cancer when dysregulated. Although lncRNAs have oncogenic or tumor-suppressive properties in lung cancer and can serve as stable biomarkers, this is still an understudied field. Here, we discuss recent evidence for lncRNAs role in lung cancer development, therapy resistance, biomarker potential, and therapeutic strategies. We conclude that understanding the interplay between non-coding and coding molecules in lung cancer should be explored further and may open up new avenues for treatment.

**Abstract:**

Long non-coding RNAs (LncRNAs) are mRNA-like molecules that do not encode for proteins and that are longer than 200 nucleotides. LncRNAs play important biological roles in normal cell physiology and organism development. Therefore, deregulation of their activities is involved in disease processes such as cancer. Lung cancer is the leading cause of cancer-related deaths due to late stage at diagnosis, distant metastasis, and high rates of therapeutic failure. LncRNAs are emerging as important molecules in lung cancer for their oncogenic or tumor-suppressive functions. LncRNAs are highly stable in circulation, presenting an opportunity for use as non-invasive and early-stage cancer diagnostic tools. Here, we summarize the latest works providing in vivo evidence available for lncRNAs role in cancer development, therapy-induced resistance, and their potential as biomarkers for diagnosis and prognosis, with a focus on lung cancer. Additionally, we discuss current therapeutic approaches to target lncRNAs. The evidence discussed here strongly suggests that investigation of lncRNAs in lung cancer in addition to protein-coding genes will provide a holistic view of molecular mechanisms of cancer initiation, development, and progression, and could open up a new avenue for cancer treatment.

## 1. Introduction

Long non-coding RNAs (LncRNAs) are broadly defined as RNAs that usually do not encode for proteins and that are longer than 200 nucleotides. These are messenger RNA (mRNA)-like molecules that are transcribed by polymerase II, 5′ capped, and have a 3′ poly-A tail. Because they can form complex secondary structures, they often have functions. Many lncRNAs are preferentially found in the nucleus, where they participate in the regulation of chromatin organization and transcription, often by forming lncRNA-DNA triplex [1,2,3], as well as through the formation of nuclear speckles and regulation of splicing. In the cytoplasm, lncRNAs can regulate mRNA stability, bind to other non-coding RNAs, and modulate protein post-translational modifications and protein function [4,5,6,7,8].

LncRNAs have been studied in mammals since the early 1990s due to their involvement in developmental processes. For example, the *Xist* (X-inactive specific transcript) lncRNA contributes to reshaping the architecture of chromatin to achieve X chromosome-silencing in early embryonic development [9]. The *H19* lncRNA is involved in genomic imprinting and regulation of the insulin growth factor 2 (*IGF2*) and other genes involved in embryonic growth [10,11]. The study of *HOX* genes, master regulators of embryonic development, led to the discovery of the lncRNA *HOTAIR* (Homeobox transcript antisense intergenic RNA). *HOTAIR* is transcribed from the antisense strand of the *HOXC* gene. 

*HOTAIR* has been reported to repress the transcription of the *HOXD* loci via interaction with PRC2 (polycomb repressive complex 2) [12], although in vivo models developed later report conflicting results regarding *HOXC* or *HOXD* genes regulation by *HOTAIR* [13,14]. Because of the importance of physiological context to understand lncRNA molecular function, more recently, other groups have reported exclusively in vivo approaches using animal models of lncRNA genetic ablation. For example, a comprehensive study developed 18 knock-out (KO) mouse models for less well-known lncRNAs with human orthologs. Certain lncRNAs’ expression was highly tissue-specific (such as *Fendrr*, *Manr* and *linc–Cox2* expressed mainly in lung), supporting a unique physiological role, while other lncRNAs were more ubiquitously expressed. Three of the analyzed lncRNAs were required for embryonic development (*Fendrr, Mdgt, Peril*), speaking to their fundamental functions [15]. Thus, the involvement of lncRNAs in normal cell physiology and organism development suggests that these may also control disease-related processes such as cancer.

Lung cancer is the leading cause of cancer-related mortality in the U.S., and non-small cell lung cancer (NSCLC) is the most common subtype. Lung cancer is commonly diagnosed in late stages, where patients present distant metastasis with 9% having a 5-year survival rate [16]. Although the development of targeted therapies (e.g., tyrosine kinase inhibitors, TKIs) has improved patient outcomes, their clinical efficacy is often limited by both innate and acquired resistance, permitting tumor progression and recurrence leading to poor survival rates [17]. It is imperative to better understand molecular drivers of tumorigenesis, metastasis, and therapy resistance in lung cancer to develop improved therapeutic strategies. LncRNAs are emerging as important molecules in cancer due to their oncogenic or tumor-suppressive functions. Here, we summarize the latest works providing in vivo evidence available for the role of lncRNAs in lung cancer, therapy resistance, and their potential as biomarkers.

## 2. Role of lncRNAs in Lung Cancer

Because of the role of lncRNAs in regulating a diverse array of cellular functions, deregulation of their activities is involved in cancer. Some mechanisms reported for lncRNAs functions in cancer are as follows: acting as miRNA sponges to modulate activity on their targets; interacting with histone-modifier enzymes to modulate known oncogene/tumor suppressor gene expression; interacting with transcription factors to repress/activate their transcriptional programs; acting as anti-sense molecules for tumor suppressor mRNAs, among other mechanisms [2,18]. Importantly, regulation of lncRNA expression and function in cancer follows similar principles to that of known oncogenes and tumor suppressors, as it can be mediated by DNA methylation [19], amplification or deletion [20], and mutation or SNPs of DNA sequences [21,22]. In this section, we provide examples of well-studied lncRNAs that are reported to have oncogenic or tumor-suppressive roles in lung cancer, as well as those with controversial functions.

*MALAT1* (Metastasis-Associated Lung Adenocarcinoma Transcript 1) was one of the first lncRNAs described to be associated with cancer. In 2003, Ji et al. analyzed the gene expression profile in human primary lung cancer tumors that subsequently metastasized or those that did not metastasize and compared their transcriptional signatures. They identified a Metastasis-Associated Lung Adenocarcinoma Transcript 1, named *MALAT1,* for its higher expression in primary tumors that metastasized. They also found a significant correlation between higher level of *MALAT1* expression in stage I lung cancer and worse survival outcomes [23]. Several studies since then have described *MALAT1* function in normal physiology and cancer [24,25,26]. Three independent groups developed an in vivo approach to describing *MALAT1* physiological function. They found that *Malat1* is highly abundant in several mouse tissues and highly conserved across species. Genetic perturbation of the *Malat1* locus in mice (using genetic deletion [27,28] or genetic inactivation approaches [29]) did not alter animal development, nuclear speckle formation, splicing, or mRNA stability. However, they described a role for *Malat1* in controlling neighboring genes expression in a tissue-specific manner that was not consistent between the three studies, especially that of *Neat1*, another lncRNA [28,29]. In the context of lung cancer, *MALAT1*-silencing did not show effects on lung cancer cell proliferation or viability in vitro [27]. To further understand the role of *MALAT1* in lung cancer metastasis, Gutschner et al. implanted EBC-1 lung cancer cells into nude mice and treated them with subcutaneous administration of an anti-sense oligonucleotide (ASO) targeting *MALAT1*. After five weeks of treatment, all primary tumors were excised. Metastasis nodules were analyzed at 12 weeks, indicating fewer and smaller metastatic nodules in the treated group, suggesting a role for *MALAT1* in promoting metastasis. Through additional in vitro studies, they report *MALAT1* inhibition results in aberrant expression of metastasis-associated genes in cell lines [30]. Although these studies support a role for *MALAT1* in promoting metastasis and regulating certain genes expression, other evidence exists for different roles. Kim et al. describe an elegant study that challenges previously reported roles for *MALAT1* in metastasis. Using the *Malat1* knock-out (KO) mouse model developed by Nakagawa et al. (LacZ and Poly-A sequences were used as transcriptional terminators inserted 69 bp downstream of the transcription start site of *Malat1* without deletion of the DNA sequence), Kim et al. crossed these mice with a breast cancer model driven by MMTV-PyMT that mimics human disease. Surprisingly, they found a 7.2-fold increase in metastatic foci and 31-fold increase in the percent of lung areas with metastatic lesions in *Malat1*-KO mice as compared to *Malat1* WT mice, suggesting a role for *Malat1* in suppressing breast cancer metastasis to the lung. This phenotype was rescued by transgenic expression of *Malat1,* suggesting that the RNA product itself diminished metastasis. Additionally, they identified *Malat1* interaction with TEAD (transcriptional enhanced associate domain) proteins in mouse-derived tumors and cell lines, which suppressed TEAD-YAP interaction and, therefore, inhibited their pro-metastatic transcriptional program [31]. This study and others highlight the importance of context (lung model vs. breast model) as well as experimental methodology to approach lncRNAs’ functional characterization (such as lncRNA genomic DNA loss vs. RNA loss reviewed in detail elsewhere [24]). Of note, DNA elements themselves within a lncRNA locus may be responsible for regulatory functions that are independent from transcript function [32,33,34,35]. In summary, *MALAT1* promotes metastasis in lung cancer, but may show opposite functions in different types of cancer depending on cellular context.

The *GAS5* (growth arrest-specific 5) gene was first described as a G0-specific gene that is inhibited by serum and growth factors [36]. In vivo, *Gas5* genetic deletion (*Gas5*^+/−^) in mice decreased bone mass and impaired bone repair, leading to osteoporosis. Mechanistically, *Gas5* positively influenced proper cell differentiation through interaction with UPF1 (a DNA/RNA helicase) to accelerate *SMAD7* mRNA decay [37]. In lung cancer, *GAS5* was found down-regulated in 72 NSCLC tumor samples as compared to their paired adjacent normal tissues, suggesting a tumor-suppressive role. Additionally, low *GAS5* expression was correlated with larger tumor size, lower differentiation levels, and higher staging of tumor–node metastasis [38]. A xenograft model of *GAS5* overexpression (OE) showed that *GAS5* OE markedly decreased tumor size compared with control [38]. Although this study did not explore a mechanism for *GAS5*-mediated tumor suppression, other studies implicate a role for *GAS5* as a miRNA sponge to negatively influence cell cycle activator genes [39] or positively influence *PTEN* levels [40]. Additionally, a recent study revealed *GAS5* is partially localized to the mitochondria where it modulates energy homeostasis by promoting the de-acetylation of malate dehydrogenase, suppressing breast cancer [41]. Taken together, a role for *GAS5* in halting the cell cycle as well as promoting cell differentiation in normal cells supports its tumor-suppressive role reported in lung cancer. Besides *GAS5*, other lncRNAs have been studied for their tumor-suppressive functions, such as *MEG3* and *TUG1*, reviewed elsewhere [42].

*LUCAT1* (Lung Cancer-Associated Transcript 1), first identified as smoke-induced and cancer-associated lncRNA1 (*SCAL1*) [43], has higher expression in lung cancer as compared to normal controls, and is also found to be overexpressed in several cancer types [44]. To our knowledge, a *LUCAT1*-deficient mouse model has not been reported. Additionally, patients with tumors that express high levels of *LUCAT1* showed poorer overall survival as compared to those with lower *LUCAT1* expression. Moreover, high *LUCAT1* levels were associated with late staging in tumor–lymph node metastasis and higher tumor volume. In NSCLC cell lines A549 and SPC-A1, *LUCAT1* modulates *p21* and *p57* expression by promoting loci methylation through PRC2 [45].

More recently, a role for *LUCAT1* in regulating immune responses has been described. *LUCAT1* genetic deletion in myeloid cells is found to enhance interferon-mediated gene transcription. *LUCAT1* acts as an immune suppressor by interacting with STAT1 and chromatin in the nucleus. It may also act by inhibiting NF-kB functions [46]. These findings suggest a tumor-promoting role for *LUCAT1* that is tumor-cell-intrinsic, in addition to a potential non-cell autonomous mechanism via the inhibition of immune surveillance, although this mechanism remains to be explored in the lung cancer context.

*HOTAIR* has been vastly studied in cancer contexts [47,48]. In NSCLC, tumor samples and cell lines expressed higher *HOTAIR* levels as compared to normal counterparts [49]. Additionally, high *HOTAIR* levels correlated with higher tumor grade and presence of lymph node metastases [49,50]. In vitro, *HOTAIR* has been reported as a direct target of the hypoxia-inducible-factor-1α (HIF-1α), therefore enhancing A549 NSCLC cells’ proliferation, migration, and invasion [51]. In vivo, tail vein injections of SPC-A1 cells with or without siRNA targeting *HOTAIR* showed that the knock-down condition reduced the number of metastasis nodules found in the lungs of immunocompromised mice [49]. *HOTAIR*-silencing resulted in a decrease in matrix metalloproteinases (MMPs, which promote invasion and migration) expression and an increase in HOXA5 levels (a tumor suppressor) in cell lines, suggesting *HOTAIR* acts through the regulation of expression of cancer-related genes [49]. Although this xenograft assay does not account for all steps required for a tumor cell to achieve metastatic colonization (extravasation, survival in blood, seeding of new site, proliferation in new site), and they measured colonization of lungs using a lung cancer cell line (same tissue), it raises the possibility that *HOTAIR* may be involved in the seeding and survival of cancer cells. Additionally, the absence of a competent immune system challenges interpretation of these results. Development of a *HOTAIR* transgenic mouse model to understand in vivo implications in lung cancer initiation and progression is necessary, similar to a *HOTAIR* inducible system recently reported for breast cancer [52]. This model showed that sustained *HOTAIR* overexpression promotes breast cancer metastasis to lungs. Overall, with the data available, *HOTAIR* seems to play an oncogenic role in lung cancer; however, robust mechanisms through which this lncRNA function remain to be uncovered.

Most studies focus on the contribution of a single lncRNA to cancer phenotypes. However, whether the lncRNAs described above are expressed simultaneously in tumors with unique or redundant functions remains to be explored in depth. For example, Esposito et al. showed that at least 80 oncogenic lncRNAs are active in NSCLC through a lncRNA-focused CRISPR screen. By further dissecting the role of two candidate lncRNAs, *CHiLL1* and *GCAWKR,* they showed these have distinct cellular localization and non-overlapping targets. Importantly, ASOs targeting both these lncRNAs yielded additive effects, suggesting that they have cooperating functions in NSCLC progression [53]. LncRNAs are generally expressed at lower levels than protein-coding genes [54]. Because of this, we speculate that lncRNAs with redundant functions may be expressed simultaneously to compensate for a higher expression of their targets in disease conditions. By examining available TCGA lung adenocarcinoma datasets containing mRNA expression data, we did not find significant correlations (negative or positive) among the expression of lncRNAs described here. However, such an analysis in combination with functional studies could shed light on mutual exclusivity relationships between certain lncRNAs. Additionally, whether certain lncRNAs are predominantly expressed at different stages of tumor progression remains to be explored. A new online resource, lncRNAfunc https://ccsm.uth.edu/lncRNAfunc (accessed on 7 May 2023), provides insights on differentially expressed lncRNAs across different cancer types and stages available in TCGA, as well as functional predictions [55]. Although this analysis did not detect any correlations between the lncRNAs mentioned here and lung cancer stages, possibly due to lack of sufficient sample sizes, these lncRNAs did show correlation with stage in other cancers; for example, *LUCAT1* was correlated with cancer stage in kidney cancer.

In summary, lncRNAs have oncogenic and tumor-suppressive roles in lung cancer, illustrated in Figure 1. LncRNAs interact with protein-coding molecules, resulting in the activation or inactivation of specific signaling pathways in cancer cells. We speculate more lung-cancer-specific lncRNAs will be identified with genome-wide transcriptomic studies.

## 3. LncRNAs in Lung Cancer Therapy Resistance

The role of non-coding RNAs in resistance to cancer therapies has been documented [48,56,57]. Mechanistically, lncRNAs can contribute to therapy resistance by promoting cell survival pathways (including autophagy, DNA repair), inhibiting apoptosis and cell cycle checkpoints, increasing self-renewal capacity and epithelial to mesenchymal transition (EMT), modulating the tumor microenvironment as well as the cellular xenobiotic stress response (drug efflux mechanisms), among others [48,58]. Here, we provide evidence for the roles of the lncRNAs discussed above in resistance to chemotherapy, radiotherapy, and targeted therapy in lung cancer. Additionally, we discuss new advances in understanding the role of lncRNAs in immune checkpoint therapy in lung cancer.

### 3.1. Role of lncRNAs in Resistance to Chemotherapy, Radiotherapy, and Targeted Therapy in Lung Cancer

In a tumor xenograft model, silencing of *MALAT1* in cisplatin resistant A549 cells led to decreased growth in nude mice (subsequently treated with cisplatin), as compared to non-targeting control. Similarly, overexpression of *MALAT1* in cisplatin-sensitive A549 lung cancer cells increased tumor volume as compared to empty vector control [59]. These data suggest a role for *MALAT1* in promoting cisplatin resistance in lung cancer. The authors suggest that modulation of STAT-3 function by *MALAT1* drives this phenotype, although a direct interaction was not confirmed [59]. While additional lung-cancer-focused studies are lacking, the *MALAT1* role in cisplatin resistance was also found in a xenograft model of oral squamous cell carcinoma [60]. In radiotherapy, *MALAT1* also promotes resistance, although this function has not been explored in lung cancer. In a xenograft model of esophageal squamous cell carcinoma, *MALAT1* levels were found to be reduced upon radiation in tumors that respond to treatment. Additionally, overexpressing *MALAT1* in xenografts did not decrease their size upon radiation exposure, while controls showed regression [61]. Furthermore, in colorectal carcinoma cell lines, *MALAT1* knockdown enhanced radiosensitivity [62]. Therefore, *MALAT1* can impact sensitivity to radiation therapy in cancer. To our knowledge, there are no robust in vivo studies addressing the role of *MALAT1* in lung cancer targeted therapy resistance (TKIs), only those in cell lines. Cheng et al. characterized differentially expressed lncRNAs in gefitinib (EGFR TKI)-sensitive PC9 cells and gefitinib-resistant PC9 cells. They found *H19* and *BC200* lncRNAs to be up-regulated in resistant cells vs. sensitive ones, while *MALAT1* and *HOTAIR* were down-regulated in the resistant setting. These data suggest that *MALAT1* may promote sensitivity to targeted therapy in lung cancer cells [63]. However, opposite roles for *MALAT1* in targeted therapy resistance in other cancers have been described. For example, *MALAT1* is overexpressed in Sutinib-resistant renal cell carcinoma tumors vs. sensitive tumors [64]. These findings suggest *MALAT1* plays a role in therapy resistance that can be highly context-specific in regard to the type of therapy (cisplatin vs. targeted therapy) or cancer primary site (lung vs. kidney) and support the need to study these functions and mechanisms in physiologically relevant settings.

*GAS5*, a tumor-suppressive lncRNA, plays a role in the sensitization of lung cancer cells to therapy. In cisplatin-resistant A549 and H1299 cells, *GAS5* overexpression reduced IC_50_ (half-maximal inhibitory concentration) values to cisplatin. In vivo, cisplatin-resistant A549 cells stably overexpressing *GAS5* yielded lower tumor volumes when injected into nude mice as compared to vector controls [65]. Additionally, a role for *GAS5* has been reported in sensitivity to targeted therapy (gefitinib, EGFR TKI). A xenograft mouse model of *GAS5* overexpression (OE), *GAS5* OE plus gefitinib, gefitinib alone, or vehicle, showed that *GAS5* OE plus gefitinib yielded the best tumor size reduction outcomes. This suggests that *GAS5* can synergize with targeted therapy to achieve better clinical outcomes [38]. Lastly, roles for *GAS5* in sensitizing lung cancer cells to radiotherapy have also been reported [66]. This evidence suggests *GAS5* as a promising target to sensitize lung tumors to cancer therapy.

*LUCAT1* can contribute to cisplatin resistance in NSCLC. Shen et al. describe a role of *LUCAT1* as sponge of miR-514-3p, whose target is ULK1, a protein involved in autophagy. Therefore, *LUCAT1* promotes cisplatin resistance by modulating autophagy [67]. Although further investigation in the context of lung cancer is limited, *LUCAT1* is known to promote resistance to DNA-damaging agents in colorectal carcinomas [68]. Because *LUCAT1* has been recently reported to play a role in immune cell regulation [46,69], its role in immune checkpoint blockade therapies should be explored in detail as a possible target for combination therapy.

The role of *HOTAIR* in therapy resistance, similar to findings described for *MALAT1*, can be complex. For example, *HOTAIR* was found in higher levels in cisplatin-resistant NSCLC tumors as compared to sensitive ones [50]. In radiotherapy, *HOTAIR* can promote resistance to radiation therapy through the inhibition of *p21* in cervical cancer [70], it can modulate β-catenin signaling in Lewis lung cancer tumors [71], and it modulates Akt signaling in breast cancer cell lines [72]. In contrast, in targeted therapy, *HOTAIR* was reported down-regulated in tumors derived from acquired and primary resistant states to EGFR-TKIs as compared to treatment naïve tumors. Here, higher levels of *HOTAIR* expression were correlated with better survival outcomes [73]. Consistent with these findings, *HOTAIR* was down-regulated in gefinitib-resistant PC9 cells vs. sensitive ones [63]. However, another study reports higher *HOTAIR* expression in gefitinib-resistant PC9 cells as compared to gefitinib-sensitive cells [74]. Lastly, in vitro assays suggest that *HOTAIR* may mediate Crizotinib (ALK/ROS1 inhibitor) resistance through the up-regulation of autophagy [75]. Taken together, *HOTAIR* promotes resistance to cisplatin and radiotherapy in lung cancer. In targeted therapy, the type of drug and differences in experimental methodologies employed may account for the confounding role of *HOTAIR*.

Programmed cell death is crucial in attaining effective therapeutic responses regardless of the type of therapy employed. LncRNAs play a significant role in modulating cell death through several mechanisms, influencing therapeutic failure and resistance. For instance, certain lncRNAs can inhibit pro-apoptotic proteins such as P53, BAX and PARP-1, while others can promote anti-apoptotic proteins such as BCL-2 directly or through activating MYC [76,77,78]. Both mechanisms lead to the inhibition of apoptosis, contributing to therapy resistance. Ferroptosis is another type of programmed cell death that uses iron-dependent accumulation of reactive oxygen species (ROS) to induce death. LncRNAs can interfere with ferroptosis by modifying levels of key proteins, such as inhibiting ACSL4 or increasing GPX4 [79]. Pyroptosis is a type of cell death that triggers an inflammatory response and its role in cancer is controversial. LncRNAs can also exert an impact on pyroptosis. In particular, *XIST* can sequester SMAD2 in the cytoplasm, impeding the transcription of *NLRP3*, an essential mediator of pyroptosis. This finding was associated with an increased resistance to cisplatin in NSCLC [78]. In summary, understanding the role of lncRNAs in regulating programmed cell death, including apoptosis, ferroptosis and pyroptosis, is key for shedding light on their contribution to therapy resistance and developing strategies to overcome it.

### 3.2. Role of lncRNAs in Immunotherapy Responses in Lung Cancer

The role of lncRNAs in resistance to immune checkpoint inhibitors (ICI) is emerging as a field of study in many cancers [80]. ICIs target immune inhibitory molecules such as PD-1, PD-L1, and CTLA-4 with the goal of re-activating immune surveillance and tumor-cell killing [81]. Even though these therapies have favorable outcomes in certain tumor types, an effective and durable response in lung cancer is achieved only in ~25% of cases [82]. Therefore, understanding the underlying molecular mechanisms of response or resistance to ICIs is critical to improve lung cancer outcomes.

*MALAT1* may be involved in regulating responses to immunotherapy. In a study of 113 NSCLC tumor samples, *MALAT1* expression was positively correlated with *PD-L1* mRNA as well as PD-L1 protein levels [83]. Here, the authors propose *MALAT1* acts as a sponge of *miR-200a-3p*, whose target is *PD-L1* [83]. Similarly, another study proposed *LINC01140* directly represses two miRNAs (*miR-377* and *miR-155-5p*) whose target is *PD-L1*. Therefore, *LINC01140* expression promotes *PD-L1* expression and a potential pro-tumorigenic microenvironment [84]. In a co-culture assay, *LINC01140-silencing* in lung cancer cells promoted higher IFN-γ secretion from cytokine-induced killer cells, as compared to non-targeting control. In a xenograft model, lung cancer cells with knock-down of *LINC01140* were injected into immunocompromised mice and received peritumoral administration of cytokine-induced killer cells upon tumor establishment. Further tumor growth was inhibited in the knock-down condition as compared to non-targeting controls. Importantly, higher levels of pro-inflammatory cytokines were found in the circulation of mice injected with *LINC01140* knock-down tumor cells compared to controls [84]. Moreover, a recent study reports an unbiased approach to understanding lncRNAs’ relationship to the tumor immune microenvironment and prediction of response to immune checkpoint therapy in NSCLC [85]. Based on lncRNAs that were correlated with immune-checkpoint expression, and taking into account overall survival data, Zhang et al. identified a signature of ten lncRNAs that they used to separate patients into “low”- and “high”-risk groups. They analyzed immune infiltrates in tumor samples and found a significantly higher density of T-cells (CD4^+^ and CD8^+^) and dendritic cells in the low-risk group (suggesting responsiveness to immunotherapy), while macrophages were higher in the high-risk group tumors (suggesting unresponsiveness to immunotherapy) [85]. Although the gold-standards for prediction of immunotherapy response are still levels of immune-checkpoint molecules, the evidence discussed here suggests there is potential for lncRNAs to function as biomarkers to predict immunotherapy response in lung cancer, as well as to be therapeutic targets in combination with immune checkpoint inhibitors.

More recently, the cGAS/STING pathway has gained importance in modulating cancer immunotherapy responses. This pathway provides a defense against microbial pathogens and malignant cells [86]. Therefore, approaches to activate it have recently emerged, such as STING agonists. These approaches have shown to synergize with immunotherapy (anti PD-L1) and achieve better responses in cancer [87]. Importantly, lncRNAs can also regulate the cGAS/STING pathway. In NSCLC, the lncRNA *PCAT1* was reported to activate the transcription of SOX2, therefore inhibiting cGAS/STING-dependent interferon responses and causing immunosuppression [88]. In non-cancer contexts, *MALAT1* has been reported to activate cGAS/STING through CREB, therefore promoting inflammatory lung conditions [89]. It is imperative to continue researching the role of lncRNAs in modulating cGAS/STING as their de-regulation may impact cancer patient responses to immunotherapy.

In summary, the lncRNAs discussed above have all been reported to modulate resistance to cancer therapies (illustrated in Figure 2), although evidence for some therapy types is limited to other cancer settings. A caveat in the data presented is the focus on association studies, in vitro and xenograft assays (immunocompromised mice) without strong mechanistic insights.

## 4. LncRNAs as Biomarkers in NSCLC

LncRNAs are highly stable molecules that can be found in the systemic circulation. They are resistant to degradation due in part to their secondary structures, transport by exosomes, and stabilizing post-translational modifications [90,91]. Therefore, the study of lncRNAs in circulation is a plausible non-invasive method of detecting and following cancer progression. Currently, the most commonly used biomarkers for NSCLC diagnosis from circulation are carcinoembryonic antigen (CEA), cytokeratin-19 fragment (CYFRA21-1), squamous cell carcinoma antigen (SCCA), prolactin (PRL), and carbohydrate antigen 125 (CA125), which can be used individually or combined as a signature [92,93]. Additionally, lncRNAs can also be stably found in urine [94] and even in nasal mucosa [95], although the latter has not been explored in the context of cancer. In this section, we review evidence for the potential of lncRNAs as disease biomarkers in lung cancer.

Exosomes are small vesicles that facilitate the transfer of cargo from one cell to another. One of the first lncRNAs found in exosomes was *PARTICLE*. It was observed not only in exosomes isolated from the MDA-MB-361 metastatic breast adenocarcinoma cell line but also in plasma samples obtained from patients exposed to radiation [2]. In lung cancer, exosomal lncRNAs have been found to play a significant role in cancer development and metastasis. For example, the exosomal lncRNA *UFC1* was demonstrated to enhance lung cancer cell proliferation and metastasis by interacting with EZH2 and subsequently reducing *PTEN* expression [96]. Due to their packaging into exosomes, lncRNAs exhibit increased stability and can travel to distant sites, having significant potential effects in biological processes of both cancer and non-cancer cells. For instance, as reported for miRNAs, exosomal lncRNAs may sustain proliferation of cancer cells at the same time that they modify pre-metastatic niches by remodeling the microenvironment towards tumor-promoting immune cells or fibroblasts, as well as increasing vessel leakiness and angiogenesis [97]. A specific example is exosomal *HOTAIRM1,* which has been observed to modulate the expression of SPON2, an extracellular matrix protein, in cancer-associated fibroblasts. This modulation promotes cancer cell migration and invasion [98]. Additionally, *PCAT6* derived from NSCLC exosomes influences macrophages polarization, promoting the shift towards M2 phenotypes that support tumor growth [99]. These findings highlight the pivotal role of exosomal lncRNAs in intercellular communication and their implications in cancer progression.

Tang et al. analyzed lncRNAs expression in the blood from 232 patients diagnosed with NSCLC as compared to healthy controls. Expression levels of three lncRNAs (*RP11-397D12.4*, *AC007403.1*, *ERICH1-AS1*) were higher in disease versus health states. Importantly, this expression pattern was stable even after freeze–thaw cycles [100]. In a similar study, the lncRNAs *SPRY4-IT1, ANRIL*, and *NEAT1* were found overexpressed in NSCLC versus healthy controls (N = 50/group), and were detected stably after several freeze–thaw cycles and after the samples’ exposure to room temperature for up to 24 h [101]. A more recent study analyzed exosomal lncRNAs from the blood of NSCLC patients vs. tuberculosis patients and healthy controls. They report higher levels of *RP5-977B1* in NSCLC compared to the two non-cancer groups. The diagnostic power of this lncRNA was greater than that of conventional biomarkers such as CEA and CYFRA21-1, and additionally worked for early-stage NSCLC, speaking to the promise of lncRNAs to detect early disease [102]. *HOTAIR* has also been evaluated for its diagnostic value for pathological staging of NSCLC. It was determined to have a power similar to that of biomarkers CEA and CYFRA21-1 [103]. *ESCCAL-1* is an oncogenic lncRNA, initially identified in esophageal cancer [104]. The results from a large cohort lung cancer screen show that *ESCCAL-1* has increased expression in serum samples from lung cancer patients as compared to serum from patients with benign nodules or healthy individuals. These studies support the use of elevated lncRNAs expression individually or as signatures as biomarkers to predict disease or as staging tools in NSCLC. Clinical trials that evaluate the use of lncRNAs as biomarkers are currently ongoing for other cancers, according to the ClinicalTrials.gov database (accessed on 12 April 2023).

Comparable to tumor-promoting lncRNAs, tumor-suppressive lncRNAs such as *GAS5* also show potential as biomarkers in NSCLC. Liang et al. compared *GAS5* levels in the plasma of 90 NSCLC patients vs. 33 healthy controls. They detected significantly lower levels of *GAS5* in plasma derived from cancer patients. Additionally, they measured the dynamics of *GAS5* before and after surgery, detecting an increase in *GAS5* levels seven days after the patients had surgery [105]. As discussed above, *GAS5* has a clear tumor-suppressive role in lung cancer; therefore, these studies are consistent and support the use of *GAS5* as biomarker for diagnosis and for responses to clinical intervention.

## 5. LncRNAs as Therapeutic Targets

The main approaches to targeting lncRNAs are similar to protein-coding genes: inhibiting oncogenic lncRNAs or restoring the function of tumor-suppressive lncRNAs. Here, we briefly describe advances in lncRNA therapeutics as well as some challenges.

Therapeutic approaches to target lncRNAs with oncogenic functions mainly use ASOs (anti-sense oligonucleotides). The core mechanisms of action of ASOs involve promoting the cleavage of their RNA targets, impeding their translation, or modulating splicing [106]. There are several types of ASOs depending on their chemical modifications, such as locked nucleic acids and morpholinos, that may make them more resistant to degradation, provide better cellular availability, and lower off-target events [106]. As mentioned in the sections above, ASOs are a strategy widely used in cell line and animal-based assays to elucidate lncRNAs role in cancer [30]. Although ASOs that target miRNAs have entered clinical trials [7,107], to our knowledge, there are no therapies in clinical trials targeting lncRNAs directly. Small molecules are another strategy to interfere with oncogenic lncRNAs function [108]. For example, quercetin, a recently developed small molecule, binds to a *MALAT1* triplex and modulates its transcript levels and functions in vitro [3]. Another small molecule recently identified, AC1NOD4Q, blocks the interaction between *HOTAIR* and EZH2 (a PRC2 subunit), impeding methylation of downstream targets [109]. Moreover, emerging strategies to silence lncRNAs in pre-clinical models use CRISPR-based approaches, although these have not reached the clinical setting [110].

Restoring the function of tumor-suppressive lncRNAs can be achieved via gene therapy or administration of synthetic RNA molecules approaches. Although gene therapy has not reached the clinic for lncRNAs as targets, there are pre-clinical models exploring it in non-cancer contexts. For example, the lncRNA *LeXis* has been explored for exogenous administration using an adeno-associated virus vector in a murine model of familial hypercholesterolemia [111].

A main limitation in translating lncRNA targeting approaches into the clinic is their relatively poor sequence conservation between humans and other species [112]. However, it is noteworthy that *MALAT1* sequence exhibits significant conservation among various species [113], rendering it a compelling candidate for in-depth investigation into its potential as a therapeutic target. Furthermore, it is worth noting that lncRNAs are highly tissue-specific thereby minimizing the likelihood of off-target effects when employed for therapeutic purposes. Several pre-clinical models have been used to explore *MALAT1* targeting, including genetically engineered mouse models, patient-derived cell lines, organoids, and xenografts. For instance, in a MMTV-PyMT breast cancer mouse model, subcutaneous delivery of *Malat1* ASO led to metastasis reduction and higher levels of cancer cell differentiation compared to non-targeting controls. Similarly, in a 3D breast cancer organoid model, *Malat1* ASOs inhibited branching morphogenesis [114]. In lung cancer, systemic administration of *Malat1* ASO in nude mice yielded a marked reduction in the colonization of patient-derived lung cancer cells in the lungs, as compared to non-targeting controls [30].

A more recent approach to targeting lncRNAs is the use of nanoparticles containing siRNA against a specific oncogenic lncRNA or nanoparticle-conjugated tumor-suppressive lncRNAs. Nanoconjugates of the tumor suppressor *MEG3* delivered intrahepatically to animals with liver cancer showed improvement in histopathology and tumor-associated biomarkers that was superior to that achieved with unconjugated *MEG3*. Although this study did not analyze differences in blood retention or side effects, it is important to consider these factors when assessing the superiority of nanoparticles as therapeutic approaches [115]. In another study, siRNA targeting *DANCR*, a tumor-promoting lncRNA, packed into nanoparticles was delivered systemically into breast cancer-bearing nude mice. This approach was effective in suppressing tumor progression with no significant changes in animal body weight and in morphology and histology of liver, kidneys, and lungs [116]. Although this strategy has not been used in in vivo models of lung cancer, its use in cell lines shows promising results as it was effective in silencing *DANCR* and reducing migration and invasion in vitro, supporting its further exploration in animal models [117].

There are several other challenges for RNA-based therapies reported from the field of miRNAs. For example, in the phase-I clinical trial for MRX34, a double-stranded *miR-34a* mimic encapsulated in liposomal nanoparticles, serious adverse events were immune-mediated toxicities, such as cytokine release syndrome, which resulted in four patient deaths and caused the trial to terminate early [118]. The delivery of these therapies is another challenge, as they may accumulate in detoxifying organs such as kidneys and liver, causing associated toxicities [119,120]. Additionally, off-target events and on-target events in non-tumor tissues can account for toxicities. Therefore, there is a need to detect immune-related toxicities in pre-clinical studies and design strategies to reduce their prevalence in order to engineer delivery strategies that target organs of interest with lower off-site effects, as well as to increase target molecule specificity.

## 6. Conclusions/Perspectives

In summary, lncRNAs regulate key biological pathways of lung cancer such as tumor development, metastasis, and resistance to current therapies, summarized in Figure 1 and Table 1. Although the lncRNAs described in this review have been studied for at least 10 years, several aspects (such as roles in immunotherapy responses) remain to be explored in the context of lung cancer (Table 1). Because of the ability of lncRNAs to modulate function of other biomolecules, a comprehensive approach to studying their role in signaling pathways is necessary and should take into account the interactome between coding and non-coding molecules. Such approach has been recently reported for neural cell differentiation processes using CRISPRi and single-cell RNA-seq approaches [121]. Additionally, with the increasing access to single-cell sequencing technologies, it would be important to question cell of origin of lncRNAs expression within a tumor, such as for *LUCAT1*, reported to modulate immune responses.

LncRNAs have highly context-dependent roles. Therefore, robust in vivo studies to dissect mechanisms that account for physiological functions of lncRNAs are essential. Additionally, studies should be conducted in the presence of a competent immune system. Assays relying on xenografts (using immunocompromised mice) have two important caveats: (1) they do not account for potential physical interactions between tumor cells and microenvironment that may alter outcomes; and (2) they fail to predict potential side effects of lncRNA-targeting therapies that are immune-system-dependent. This gap in knowledge presents an opportunity for the field to develop such approaches.

Lastly, even though the role of lncRNAs in therapy resistance is documented, a role in the persister cell state (minimal residual disease) has not been explored. Such studies could shed light on important survival mechanisms that drive therapeutic failure and disease recurrence. LncRNAs show promise as disease biomarkers based on their highly stable nature in circulation. An important aspect to explore deeper is their potential to detect disease in early stages and temporal circular detection of lncRNAs to monitor the therapeutic response. Such an approach could be implemented for routine surveillance in advanced cancer patient groups.

## Figures and Tables

**Figure 1 cancers-15-03135-f001:**
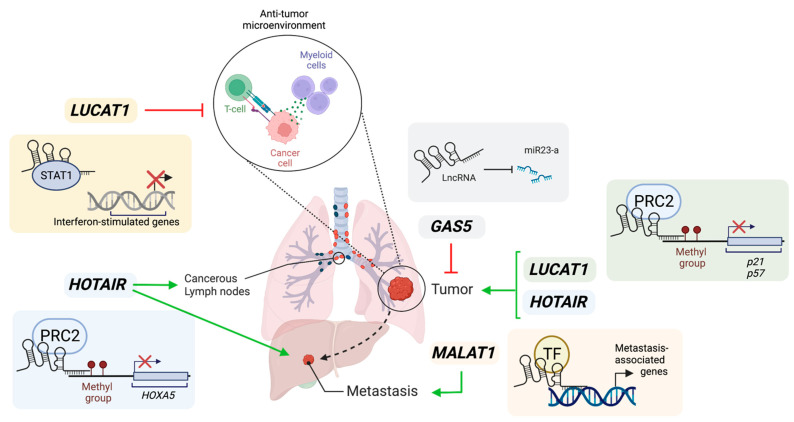
Illustration of roles of lncRNAs in lung cancer. *GAS5* acts as a miRNA sponge to activate their mRNA targets and inhibit cancer cell proliferation, therefore acting as a tumor suppressor. *LUCAT1* can promote cancer cell proliferation via epigenetic silencing of *p21* and *p57* loci. Additionally, *LUCAT1* may create a pro-tumorigenic microenvironment by inhibiting interferon-mediated responses through the sequestration of STAT1. *HOTAIR* can promote cancer cell proliferation and metastasis via the recruitment of PRC2 to methylate loci and repress gene expression, such as *HOXA5* which is a tumor suppressor. *MALAT1* stimulates lung cancer metastasis by potential recruitment of transcription factors (TF) to promoters of metastasis-associated genes. Green arrows represent positive influence. Red arrows represent inhibition. Created with Biorender.com (accessed on 4 June 2023).

**Figure 2 cancers-15-03135-f002:**
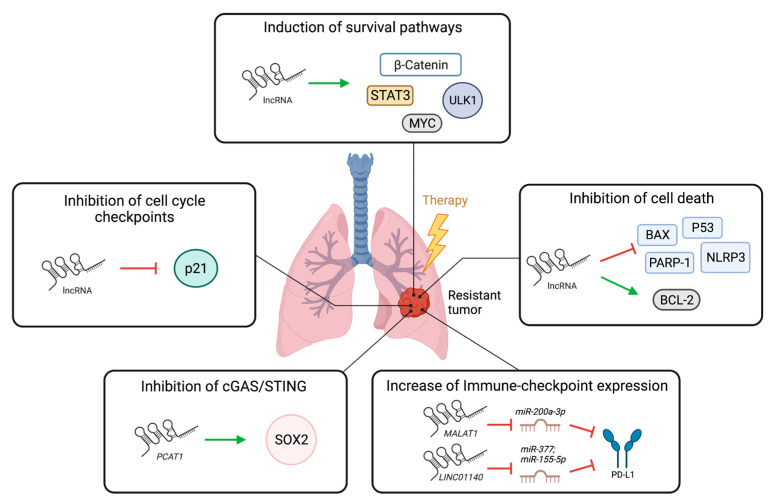
Illustration of lncRNAs contribution to therapy resistance. LncRNAs can induce survival pathways through activation of several proteins such as β-catenin, STAT3, MYC, or ULK1. LncRNAs inhibit cell death by inducing anti-apoptotic proteins such as BCL-2 or inhibiting pro-apoptotic proteins such as P53, BAX, PARP-1, and NLRP3. LncRNAs promote increase in PD-L1 levels by inhibiting miRNAs that target PD-L1. *PCAT1* induces *SOX2* to inhibit cGAS/STING pathway and diminish interferon responses, thereby modulating immune-therapy efficacy. LncRNAs may inhibit cell cycle checkpoints by modulating *p21*, although this has not been explored in the context of lung cancer. Green arrows represent induction, red arrows represent inhibition. Created with Biorender.com (accessed on 4 June 2023).

**Table 1 cancers-15-03135-t001:** Summary of roles of lncRNAs *GAS5, MALAT1, LUCAT1*, and *HOTAIR* in normal physiology and in lung cancer.

LncRNA	Normal Function	Role in Lung Cancer	Role in Chemotherapy	Role in Radiotherapy	Role in Targeted Therapy	Role in Immunotherapy	Potential as Biomarker
			In Lung Cancer				
*GAS5*	Cell cycle inhibition [36], cell differentiation [37]	Tumor suppressor [38]	Promotes sensitivity to cisplatin [65]	Promotes sensitivity [66]	Promotes sensitivity to EGFR TKI [38]	Not explored	Yes—lower levels in NSCLC [105]
*MALAT1*	Regulation of neighboring genes expression [28,29]	Oncogene [23,30]	Promotes resistance to cisplatin [59]	Not explored	May promote sensitivity: Down-regulated in EGFR-TKI resistant PC9 cells [63]	May be associated with therapeutic failure: correlated with PD-L1 expression [83]	Yes—higher levels in NSCLC [122]
*LUCAT1*	Inhibition of immune responses [46,69]	Oncogene [43,45]	Promotes resistance to cisplatin [67]	Not explored	Not explored	Not explored	Yes—higher levels in LUAD [123]
*HOTAIR*	Regulation of *HOX* genes expression [12,13,14] by recruitment of histone-modifier enzymes [124]	Oncogene [49,50,72]	Promotes resistance to cisplatin [50]	Promotes resistance [71]	Controversial roles: Down-regulated in EGFR-TKI resistant tumors [73]; Down-regulated in EGFR-TKI resistant PC9 cells [63]; Up-regulated in EGFR-TKI resistant PC9 cells [74]; Promotes resistance to Crizotinib (ALK/ROS1 inhibitor) [75].	Not explored	Yes—higher levels in NSCLC [103]

TKI = tyrosine-kinase inhibitor, NSCLC = non-small cell lung cancer, LUAD = lung adenocarcinoma.

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
