# Peer review of "Long Non-Coding RNAs as Emerging Targets in Lung Cancer"

_cancers, 2023, doi:10.3390/cancers15123135_

Round 1
Reviewer 1 Report
In this review article, they address the function of Long non-coding RNAs (LncRNAs) in lung cancer and describe the detailed mechanism of resistance pathway and the role of immunotherapy in lung cancer. However, the article still had some concerns needed to be addressed as below:
1. In sections 2, the authors need to address the rationale to describe why choose those LncRNAs in this review article.
2. In section 3, the authors need to draw a schematic diagram to address the overall resistance mechanism. In these two sections, it will be easier to follow the descriptions if the authors have a picture to conclude previous findings.
3. Drug resistance in lung cancer is critical in clinic. And cell death such as apoptosis, ferroptosis and pyroptosis is correlated with drug resistance. Do any previous studies show LncRNA affects cell death in lung cancer?
4. Authors mentioned the importance of LncRNAs in immunotherapy. Recently, the cGAS/STING innate immune pathway has been shown to be important for the efficacy of immunotherapy. Thus, if authors can describe the role of LncRNAs in the innate immunity of lung cancer will raise the value of this article.
Author Response
Thank you for your helpful feedback. We have addressed your suggestions point by point below. All changes made to the text are tracked and in blue font.
- We have highlighted our rationale to discuss specific lncRNAs in our manuscript in the first paragraph of section 2 (page 2).
“In this section, we provide examples of well-studied lncRNAs that are reported to have oncogenic or tumor suppressive roles in lung cancer, as well as those with controversial functions”
- We have included a new figure (Figure 2) summarizing mechanisms of lncRNAs contribution to therapy resistance in section 3 (page 8).
Figure 2. Illustration of lncRNAs contribution to therapy resistance. LncRNAs can induce survival pathways through activation of several proteins such as b-catenin, STAT3, MYC, or ULK1. LncRNAs inhibit cell death by inducing anti-apoptotic proteins like BCL-2 or inhibiting pro-apoptotic proteins such as P53, BAX, PARP-1, and NLRP3. LncRNAs promote increase in PD-L1 levels by inhibiting miRNAs that target PD-L1. PCAT1 induces SOX2 to inhibit cGAS/STING pathway and diminish interferon responses, thereby modulating immune-therapy efficacy. LncRNAs may inhibit cell cycle checkpoints by modulating p21, although this has not been explored in the context of lung cancer. Green arrows represent induction, red arrows represent inhibition. Created with Biorender.com.
- We have added specific evidence for lncRNAs modulation of cell death in section 3.1 (page 7).
“Programmed cell death is crucial in attaining effective therapeutic responses regardless of the type of therapy employed. LncRNAs play a significant role in modulating cell death through several mechanisms, influencing therapeutic failure and resistance. For instance, certain lncRNAs can inhibit pro-apoptotic proteins like P53, BAX and PARP-1, while others can promote anti-apoptotic proteins such as BCL-2 directly or through activating MYC [76, 77]. Both of these mechanisms lead to the inhibition of apoptosis, contributing to therapy resistance. Ferroptosis is another type of programmed cell death that uses iron-dependent accumulation of reactive oxygen species (ROS) to induce death. LncRNAs can interfere with ferroptosis by modifying levels of key proteins, such as inhibiting ACSL4 or increasing GPX4 [78]. Pyroptosis is a type of cell death that triggers an inflammatory response and its role in cancer is controversial. LncRNAs can also exert an impact on pyroptosis. In particular, XIST can sequester SMAD2 in the cytoplasm, impeding the transcription of NLRP3, an essential mediator of pyroptosis. This finding was associated with an increased resistance to cisplatin in NSCLC[79]. In summary, understanding the role of lncRNAs in regulating programmed cell death, including apoptosis, ferroptosis and pyroptosis, is key for shedding light on their contribution to therapy resistance and developing strategies to overcome it”.
- Thank you for this important suggestion. We have added evidence for the role of lncRNAs in cGAS/STING pathway modulation in section 3.2 (page 8).
“More recently, the cGAS/STING pathway has gained importance in modulating cancer immunotherapy responses. This pathway provides defense against microbial pathogens and malignant cells [86]. Therefore, approaches to activate it have recently emerged, such as STING agonists. These approaches have shown to synergize with immunotherapy (anti PD-L1) and achieve better responses in cancer [87]. Importantly, lncRNAs can also regulate the cGAS/STING pathway. In NSCLC, the lncRNA PCAT1 was reported to activate transcription of SOX2, therefore inhibiting cGAS/STING-dependent interferon responses and causing immunosuppression [88]. In non-cancer contexts, MALAT1 has been reported to activate cGAS/STING through CREB, therefore promoting inflammatory lung conditions [89]. It is imperative to continue researching the role of lncRNAs in modulating cGAS/STING as their de-regulation may impact cancer patient responses to immunotherapy”.
Reviewer 2 Report
This review summarises the important emerging role of long non-coding RNA into the oncology field of lung cancer. Focus is mainly placed on MALAT1, GAS5, HOTAIR and LUCAT1 with other lesser known lncRNAs discussed in the context of lung cancer and cancer types.
The review is well written and makes a further contribution to the literature.
A few minor comments:
1. No keywords listed.
2. The role of triplex formation should be mentioned in the intro when discussing secondary structures (eg. role of radiation sensitive PARTICLE)
3. Please cite lncRNA PARTICLE when discussing the presence of lncRNA in exosomes as this was one of the first to be found there (Cell Reports, 2015). Also, in section 2 include along with citation 15 as it acts as a suppressor of Tumor suppressors MAT2A and WWOX.
4. Perhaps define the TEAD abbreviation.
5. A suggestion – put GAS5 at the top in the table as it is a tumor suppressor.
Author Response
Thank you very much for your feedback. We have addressed your comments point by point below. All changes made to the text are tracked and in blue font.
- We have listed key words in page 1.
- We have added lncRNAs triplex formation in the introduction (page 2).
“Many lncRNAs are preferentially found in the nucleus, where they participate in regulation of chromatin organization and transcription, often by forming lncRNA-DNA triplex [1-3]”
- We have cited the Cell Reports publication discussing PARTICLE in exosomes in section 4 (page 9). We have cited this publication as well as in section 2, together with citation 15 (new citation 18) in page 2.
“Exosomes are small vesicles that facilitate the transfer of cargo from one cell to another. One of the first lncRNAs found in exosomes was PARTICLE. It was observed not only in exosomes isolated from the MDA-MB-361 metastatic breast adenocarcinoma cell line but also in plasma samples obtained from patients exposed to radiation [2]. In lung cancer, exosomal lncRNAs have been found to play a significant role in cancer development and metastasis. For example, the exosomal lncRNA UFC1 was demonstrated to enhance lung cancer cell proliferation and metastasis by interacting with EZH2 and subsequently reducing PTEN expression [95]. Due to their packaging into exosomes, lncRNAs exhibit increased stability and can travel to distant sites, having significant potential effects in biological processes of both cancer and non-cancer cells. For instance, as reported for miRNAs, exosomal lncRNAs may sustain proliferation of cancer cells at the same time that they modify pre-metastatic niches by remodeling the microenvironment towards tumor-promoting immune cells or fibroblasts, as well as increasing vessel leakiness and angiogenesis [96]. A specific example is exosomal HOTAIRM1, which has been observed to modulate the expression of SPON2, an extracellular matrix protein, in cancer-associated fibroblasts. This modulation promotes cancer cell migration and invasion [97]. Additionally, PCAT6 derived from NSCLC exosomes influences macrophages polarization, promoting the shift towards M2 phenotypes that support tumor growth [98]. These findings highlight the pivotal role of exosomal lncRNAs in intercellular communication and their implications in cancer progression”.
- We have defined TEAD in section 2 (page 3).
- We have changed the order of lncRNAs in Table 1, now showing GAS5
Reviewer 3 Report
This review outlines the role of lncRNAs in lung cancer and their potential manipulation as a targeted therapy. It focuses on specific lncRNAs relevant to lung cancer, how lncRNAs resist both conventional and immunotherapeutic targeting, the usefulness of lncRNAs as biomarkers and briefly how to exploit oncogenic lncRNAs.
This is a very well written paper and acts as a general review for the evolving field of targeting lung cancer specific lncRNAs therapeutically.
The section describing the challenges inherent to using lncRNAs as therapeutic targets could be more thorough. Although not currently in use clinically, more detailed explanations of pre-clinical models would be informative.
This review lacks description of how lncRNAs are present in exosomes of lung cancer cells and how this contributes to metastatic niche formation.
Need to have consistency in the use of LncRNA vs lncRNA in the text, it switches back and forth.
Author Response
Thank you for your kind words and constructive suggestions. We have addressed your comments below. All changes made to the text are tracked and in blue font.
- We have added paragraphs on pre-clinical models of lncRNA targeting, in section 5 (page 10).
“A main limitation in translating lncRNA targeting approaches into the clinic is their relatively poor sequence conservation between humans and other species [111]. However, it is noteworthy that MALAT1 sequence exhibits significant conservation among various species [112], rendering it a compelling candidate for in-depth investigation into its potential as a therapeutic target. Furthermore, it is worth noting that lncRNAs are highly tissue-specific thereby minimizing the likelihood of off-target effects when employed for therapeutic purposes. Several pre-clinical models have been used to explore MALAT1 targeting, including genetically engineered mouse models, patient-derived cell lines, organoids, and xenografts. For instance, in a MMTV-PyMT breast cancer mouse model, subcutaneous delivery of Malat1 ASO led to metastasis reduction and higher levels of cancer cell differentiation compared to non-targeting controls. Similarly, in a 3D breast cancer organoid model, Malat1 ASOs inhibited branching morphogenesis [113]. In lung cancer, systemic administration of Malat1 ASO in nude mice yielded a marked reduction in the colonization of patient-derived lung cancer cells in the lungs, as compared to non-targeting controls [30].
A more recent approach to targeting lncRNAs is the use of nanoparticles containing siRNA against a specific oncogenic lncRNA or nanoparticle-conjugated tumor suppressive lncRNAs. Nanoconjugates of the tumor suppressor MEG3 delivered intrahepatically to animals with liver cancer showed improvement in histopathology and tumor-associated biomarkers that was superior to that achieved with unconjugated MEG3. Although this study did not analyze differences in blood retention or side effects, it is important to consider these factors when assessing the superiority of nanoparticles as therapeutic approaches [114]. In another study, siRNA targeting DANCR, a tumor-promoting lncRNA, packed into nanoparticles was delivered systemically into breast cancer-bearing nude mice. This approach was effective in suppressing tumor progression with no significant changes in animal body weight and in morphology and histology of liver, kidneys, and lungs[115]. Although this strategy has not been used in in vivo models in lung cancer, its use in cell lines shows promising results as it was effective in silencing DANCR and reducing migration and invasion in vitro, supporting its further exploration in animal models [116]”.
- We have added evidence for lncRNAs presence in exosomes in lung cancer cells and their contribution to cancer in section 4 (page 9).
“Exosomes are small vesicles that facilitate the transfer of cargo from one cell to another. One of the first lncRNAs found in exosomes was PARTICLE. It was observed not only in exosomes isolated from the MDA-MB-361 metastatic breast adenocarcinoma cell line but also in plasma samples obtained from patients exposed to radiation [2]. In lung cancer, exosomal lncRNAs have been found to play a significant role in cancer development and metastasis. For example, the exosomal lncRNA UFC1 was demonstrated to enhance lung cancer cell proliferation and metastasis by interacting with EZH2 and subsequently reducing PTEN expression [95]. Due to their packaging into exosomes, lncRNAs exhibit increased stability and can travel to distant sites, having significant potential effects in biological processes of both cancer and non-cancer cells. For instance, as reported for miRNAs, exosomal lncRNAs may sustain proliferation of cancer cells at the same time that they modify pre-metastatic niches by remodeling the microenvironment towards tumor-promoting immune cells or fibroblasts, as well as increasing vessel leakiness and angiogenesis [96]. A specific example is exosomal HOTAIRM1, which has been observed to modulate the expression of SPON2, an extracellular matrix protein, in cancer-associated fibroblasts. This modulation promotes cancer cell migration and invasion [97]. Additionally, PCAT6 derived from NSCLC exosomes influences macrophages polarization, promoting the shift towards M2 phenotypes that support tumor growth [98]. These findings highlight the pivotal role of exosomal lncRNAs in intercellular communication and their implications in cancer progression”.
- We consistently use “lncRNAs” now, unless the word is at the beginning of a sentence where we used “LncRNAs”.